# The Phenotypic Variability Associated with Hepatocyte Nuclear Factor 1B Genetic Defects Poses Challenges in Both Diagnosis and Therapy

**DOI:** 10.3390/ijms25084552

**Published:** 2024-04-22

**Authors:** Ioannis Petrakis, Maria Sfakiotaki, Maria Bitsori, Eleni Drosataki, Kleio Dermitzaki, Christos Pleros, Ariadni Androvitsanea, Dimitrios Samonakis, Amalia Sertedaki, Paraskevi Xekouki, Emmanouil Galanakis, Kostas Stylianou

**Affiliations:** 1Department of Nephrology, University of Crete, 71500 Heraklion, Greece; petrakgia@gmail.com (I.P.); elenidro2@hotmail.com (E.D.); ekderm@gmail.com (K.D.); xpleros@gmail.com (C.P.); ariaandrovitsanea@gmail.com (A.A.); 2Department of Endocrinology, University of Crete, 71500 Heraklion, Greece; mariasfak@yahoo.gr (M.S.); pxekouki@uoc.gr (P.X.); 3Department of Pediatrics, University of Crete, 71500 Heraklion, Greece; bitmar2@gmail.com (M.B.); emmgalan@uoc.gr (E.G.); 4Department of Gastroenterology, University of Crete, 71500 Heraklion, Greece; dsamonakis@gmail.com; 5First Department of Pediatrics, Medical School, National and Kapodistrian University of Athens, 11527 Athens, Greece; aserted@med.uoa.gr

**Keywords:** cystic kidney disease, genetic variability, hepatic nuclear factor 1B (HNF1B), multiplex ligation-dependent probe amplification (MPLA), whole exome sequencing (WES)

## Abstract

The evolving landscape of clinical genetics is becoming increasingly relevant in the field of nephrology. HNF1B-associated renal disease presents with a diverse array of renal and extrarenal manifestations, prominently featuring cystic kidney disease and diabetes mellitus. For the genetic analyses, whole exome sequencing (WES) and multiplex ligation-dependent probe amplification (MLPA) were performed. Bioinformatics analysis was performed with Ingenuity Clinical Insights software (Qiagen). The patient’s electronic record was utilized after receiving informed consent. In this report, we present seven cases of HNF1B-associated kidney disease, each featuring distinct genetic abnormalities and displaying diverse extrarenal manifestations. Over 12 years, the mean decline in eGFR averaged −2.22 ± 0.7 mL/min/1.73 m^2^. Diabetes mellitus was present in five patients, kidney dysplastic lesions in six patients, pancreatic dysplasia, hypomagnesemia and abnormal liver function tests in three patients each. This case series emphasizes the phenotypic variability and the fast decline in kidney function associated with HNF-1B-related disease. Additionally, it underscores that complex clinical presentations may have a retrospectively straightforward explanation through the use of diverse genetic analytical tools.

## 1. Introduction

Hepatocyte nuclear factor-1beta (HNF-1B) serves as a ubiquitously expressed transcription factor that is crucial for the embryonic development of the kidney, pancreas, liver, and Mullerian duct [1]. Experimental ablation of *HNF1B* results in cyst formation and aberrant renal epithelial cells TGF-β signaling, resulting in kidney fibrosis [2].

*HNF1B*-associated disease is inherited in an autosomal dominant pattern; however, many patients present de novo pathogenic variants (50% of cases) with a negative family history. Genetic anomalies associated with *HNF1B* include single-nucleotide variations, small indels, and whole gene deletions [3]. To date, over 230 distinct *HNF1B* variants have been documented [4]. The most common genetic alteration (around 50% of patients) is a complete gene deletion, in the context of 17q12 chromosomal microdeletion, which also includes at least 14 other genes [4]. This genetic diversity results in heterogeneous clinical presentations, giving rise to a spectrum of manifestations such as multicystic and/or dysplastic kidney disease, maturity-onset diabetes of the young (MODY), genital tract abnormalities, pancreatic atrophy, and aberrant liver function tests [5]. Hypomagnesemia and hypokalemia are seen in 62% and 46% of mutation carriers, respectively [6]. However, the presence of diabetes or renal cysts is not universal [4]. 

Herein, we present seven cases of *HNF1B*-associated disease exhibiting varying renal and extrarenal manifestations. Accurate genetic diagnosis necessitated a comprehensive array of diagnostic techniques, including whole exome sequencing (WES), Sanger analysis, and multiplex ligation-dependent probe amplification (MLPA). MLPA, a method adept at detecting copy number variations and medium-to-large size DNA defects, played a crucial role in identifying genetic abnormalities in some patients [7]. 

## 2. Case Presentation

A collaborative protocol was established and implemented across three outpatient clinics at our tertiary hospital: adult nephrology, pediatric nephrology, and endocrinology. The aim was to investigate the prevalence and detailed clinical and genetic features of patients suspected to have *HNF1B*-related syndrome based on the presence of multicystic/dysplastic kidney disease and/or maturity-onset diabetes of the young. This protocol was developed and approved in 2015. 

In all eligible patients attending the 3 outpatient clinics, we conducted a comprehensive set of genetic and biochemical tests to elucidate the possibility of an *HNF1B*-related disease. These tests encompass WES, confirmatory and segregation analysis via Sanger sequencing and MLPA. More specifically, genomic DNA was extracted from the specimens. Standard library kits were used to target the exon regions of the patient’s genome, and targeted regions were sequenced using the Illumina platform with 100bp paired-end reads. The DNA sequence was mapped to and analysed in comparison with the published human genome build UCSC hg38 reference sequences. After the removal of low-quality and duplicate reads, variants were identified using the GATK Haplotype Caller and Germline CNV Caller software (version 4.5.0.0). This test detects single-nucleotide variations (SNVs), small insertions and deletions (indels) in the DNA coding sequences, nearby flanking regions (+/−20bp) and known splice regions in the genes targeted as well as CNVs. The average depth of coverage was >80X, and over 99.8% of the exome-targeted region was covered. Reportable sequence variants in the proband were confirmed by di-deoxy DNA sequence analysis. Bioinformatics analysis was conducted utilizing Ingenuity Clinical Insights software version 23.0.1 from Qiagen Inc., (Aahrus, Denmark, EU) incorporating data from the Human Gene Mutation Database (HGMD). 

Clinical and laboratory data were gathered from the patients’ medical records over a 12-year period (except for one newly diagnosed patient), incorporating both retrospective and prospective information post-diagnosis. This data compilation aimed to assess the rate of decline in the estimated glomerular filtration rate (eGFR). The study protocol is in accordance with the Helsinki Declaration of 1975, as revised in 2013, and has been approved by the University Hospital of Heraklion ethical committee. 

Over a period of 8 years, we have diagnosed eight individuals with *HNF1B* gene alterations, with seven of them exhibiting pathological clinical features thus far.

Patient 1, a 24-year-old Caucasian male, presented with a gradual decline in renal function over the past decade, with the most recent estimated glomerular filtration rate (eGFR) measuring 81 mL/min/1.73 m^2^ (Figure 1A) and a relatively fast pace of eGFR decline at 3 mL/min/1.73 m^2^/year. Notably, he exhibits left kidney agenesis, accompanied by pancreas body and tail agenesis, along with hypospadias, hypomagnesemia, hypercalcemia and hyperparathyroidism (Figure 1B). At the age of 22, he developed diabetes mellitus (DM), which was treated with Dulaglutide, currently at a dose of 1.5 mg weekly. MLPA identified a heterozygous *HNF1B* gene deletion (Figure 1B), confirming that MLPA can more easily identify large deletions compared to WES.

Patient 2, the 29-year-old brother of Patient 1, shares the same *HNF1B* defect, presenting with left kidney agenesis and multiple right kidney cysts contributing to renal function deterioration (latest eGFR 79 mL/min/1.73 m^2^, eGFR decline at 3 mL/min/1.73 m^2^/year), dorsal pancreas agenesis, psoriasis, diabetes and abnormal liver function tests. He has severe hypomagnesemia with relatively low parathormone. Interestingly, their mother and grandmother have a history of early-onset DM (Figure 1B).

In an unrelated case, a 19-year-old Caucasian male (Patient 3) without a prior family history of kidney disease underwent evaluation for multicystic kidney disease and persistently elevated liver enzymes. eGFR declined at a pace of 2.1 mL/min/1.73 m^2^/year (Figure 1). WES analysis failed to detect pathogenic alterations in the relevant genes. However, MLPA identified a de novo heterozygous *HNF1B* gene deletion (with normal *HNF1B* genetic tests for the parents). Notably, no abnormalities in the pancreas or urinary tract, and no proteinuria or hypomagnesemia, were observed. The fourth case involves a 14-year-old Caucasian male diagnosed with a dysplastic right kidney at the age of four, featuring contralateral multicystic disease and calcifications in the left kidney. His pancreas was hypoplastic with a benign solitary cyst. At eight years old, he experienced an episode of diabetic ketoacidosis. WES revealed a heterozygous, likely pathogenic (PM1, PM2, PM5, and PP3) point mutation within the *HNF1B* gene (p.Asn289Thr). He displayed a slightly accelerated decline in eGFR over the last 12 years, at 1.5 mL/min/1.73 m^2^ per year (Figure 1A). His father and brother exhibited no abnormalities. His mother (Patient 5), who shared the same *HNF1B* alteration, presented a dysplastic left kidney with nephrolithiasis, eGFR decline at 2.5 mL/min/1.73m2 per year and DM during her pregnancy (Figure 1B). Patient 6, a 4-year-old Caucasian boy, was initially diagnosed with autosomal recessive polycystic kidney disease. Prenatal hyperechogenic kidneys led to that diagnosis, and post-birth ultrasounds revealed multiple small cysts in the kidneys and a normal pancreas. He is the first child in a phenotypically healthy family to experience mild psychomotor developmental delay during infancy, autistic features and cryptorchidism. Liver function tests show a consistent slight increase in SGOT and SGPT, while kidney function is compromised with the most recent estimation of GFR at 65mL/min/1.73m² and trace proteinuria. WES with CNV calling revealed the presence of 17q12 recurrent deletion syndrome (17:36486450-37745134 deletion), a large deletion of 19 genes, including the *HNF1B* gene (*AATF*, *ACACA*, *C17orf78*, *DDX52*, *DHRS11*, *DUSP14*, *GGNBP2*, *HNF1B*, *LHX1*, *LHX1-DT*, *MIR2909*, *MIR378J*, *MRM1*, *MYO19*, *PIGW*, *SNORA90*, *SYNRG*, *TADA2A*, and *ZNHIT3*). This patient was not included in the estimation of eGFR slope decline due to the lack of serial long-term measurements. 

In contrast to previous cases in which kidney disease was part of the phenotype, we describe a case of a 61-year-old woman (Patient 7) who was diagnosed with DM2 based on a glucose level of 210 mg/dL at 120’ of a 2 h oral glucose tolerance test (OGTT) and an increased HbA1c of 6.3%. The investigation was prompted by her family history, as both her parents had DM2 treated with oral antidiabetic medications. Her medical history was significant for hyperlipidaemia and ulcerative colitis. She was started on metformin with good glycaemic control. Genetic analysis with WES revealed the *HNF1B* p.His336Asp variation. 

Imaging investigation of the liver, kidneys, pancreas and spleen did not reveal any abnormalities, and kidney function was normal, with an almost normal eGFR slope of −1.1 mL/min/1.73 m^2^. The same variation was detected in one of her daughters who appears absolutely healthy so far. Table 1 summarizes the demographic, clinical, and genetic data of our series.

## 3. Discussion

Heterozygous pathogenic variants in the *HNF1B* gene appear as the predominant genetic culprits behind organ malformations in pediatric populations, giving rise to a diverse array of clinical presentations. These pertaining to kidneys include prenatal hyperechogenic kidney, chronic tubulointerstitial kidney disease, renal cystic disease, renal hypo-dysplasia, glomerulocystic kidney disease, horseshoe kidney, bilateral hydronephrosis, nephrocalcinosis, nephrogenic diabetes insipidus and hypomagnesemia due to renal magnesium wasting. Extrarenal manifestations include MODY, pancreatic hypoplasia, abnormal liver tests, gout and genital tract malformations [8]. 

The HNF1B protein plays a crucial role in regulating tissue-specific gene expression in various epithelial cells found in multiple organs, including the kidney, pancreas, liver and genitourinary tract. Additionally, the transient expression of *HNF1B* is observed in tissues such as the neural tube, epididymis, seminal vesicles, prostate and uterus. This diverse expression pattern accounts for the highly variable presentation of *HNF1B*-related disorders [8].

The most common clinical renal presentation observed is chronic tubulointerstitial nephritis, characterized by nonsignificant urinalysis and a gradual decline in renal function. Renal biopsy shows a pattern of interstitial fibrosis and tubular atrophy. Notably, a similar phenotype is often observed in patients with heterozygous variants of *UMOD*, *MUC1*, and *REN* genes, which cause an autosomal dominant form of tubulointerstitial nephritis. Hence, *HNF1B* nephropathy is now classified as part of the group of disorders termed Autosomal Dominant Tubulointerstitial Kidney Disease [8,9].

Whole-gene deletions represent a substantial proportion, accounting for up to 50% of variants, usually in the context of 17q12 chromosomal microdeletion. The deletions range from 1.2 to 1.85 Mb and may include as many as 15 genes [10]. De novo variants constitute approximately 40% of *HNF1B* variants [3] and, therefore, a family history of the disease may be absent. As of now, there is no definitive correlation established between the type or the position of a pathogenic variation within *HNF1B* and the occurrence of specific clinical features [4]. Given these genetic intricacies, the integration of clinical genetics into nephrology practices becomes imperative. In the diagnostic toolkit, MLPA emerges as a valuable tool for identifying large genomic defects in individuals with cystic kidney disease. Meanwhile, WES proves instrumental in revealing point mutations and small variations [7]. Furthermore, deep WES with CNV calling can effectively identify 17q12 deletions, the most common genetic anomaly of *HNF1B*-related disease. Notably, even with identical *HNF1B* allelic variations, our patients exhibited distinct trajectories of renal function deterioration, as evidenced by the variable slope of eGFR decline. The mean decline of eGFR averaged −2.22 +/− 0.7 mL/min/1.73 m^2^ in our series, similar to other published studies [3]. Timely recognition of extrarenal manifestations, such as diabetes mellitus and hypertension, holds the potential for initiating therapeutic interventions that can delay renal function decline. In a large French study, renal biopsy was performed in 20 patients and revealed various lesions, including interstitial fibrosis in 10 and diabetic glomerular disease in 5. The progression of kidney disease was found to be more strongly associated with the presence of hypertension, a nucleotide variation versus deletion, proteinuria and age rather than solely with the presence or absence of diabetes [9]. Therefore, all of these factors should be considered when designing our therapeutic strategy.

The p.His336Asp alteration (Patient 7) has been classified as a variant of uncertain significance (VUS) in the ClinVar database. However, in a recent cohort study from Croatia, this alteration was considered pathogenic, as it was segregated in several members of two families exhibiting MODY [11]. Similar findings were reported by a Turkish group [12]. In these two previous studies, the p.His336Asp alteration has been linked solely with MODY, devoid of any kidney manifestations, which aligns with our observations in patient 7. In contrast, an older study [13] associated this alteration with the presence of renal hypoplasia and cystic dysplasia–horseshoe kidney in two patients each. 

The diverse spectrum of manifestations associated with *HNF1B*-related renal disease often leads to diagnostic delays, contributing to significant progression in chronic kidney disease and increased disease burden. Accordingly, we emphasize the critical importance of early diagnosis in patients with *HNF1B*-related disease. 

## 4. Conclusions

Clinical presentations that initially appear complex may, upon genetic analysis, reveal a retrospectively straightforward explanation. In contemporary practice, all patients with renal cysts, genital tract malformations, hypomagnesemia, liver test abnormalities and diabetes should undergo screening for *HNF1B*-associated disease. Collaborative efforts among diverse medical disciplines are indispensable for elucidating the intricate associations between phenotype–genotype, irrespective of whether chronic kidney disease is the cardinal manifestation.

## Figures and Tables

**Figure 1 ijms-25-04552-f001:**
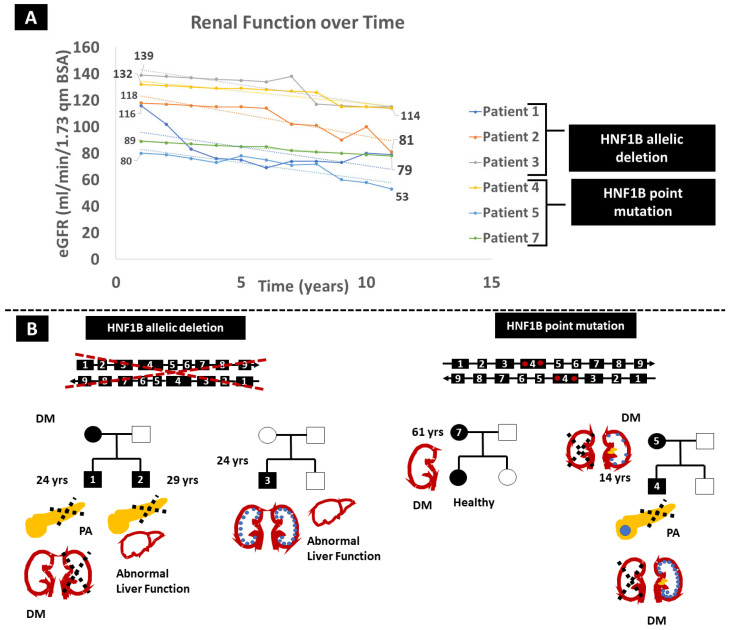
(**A**) Renal function deterioration trajectories (trends depicted as dotted lines) over time vary among various *HNF1B* genetic alterations. (**B**) Illustration of the primary clinical abnormalities associated with genetic alterations. Key annotations include diabetes mellitus (DM), pancreatic agenesis (PA), blue dots indicating renal cysts, yellow cylinders within the kidney denoting lithiasis, crossed dotted lines representing non-functional organs, red dot indicating a point mutation, a large blue circle representing a pancreatic cyst, and numerical identifiers within the family tree corresponding to patients 1, 2, 3, 4 and 5, respectively.

**Table 1 ijms-25-04552-t001:** Demographic, clinical and genetic data of 7 patients with HNF1B syndrome.

Patient	Age	Sex	Genetic Variant	eGFR Decline Rate	DM	Dysplastic Abnormalities	Low sMg	Other
1	24	Μ	*HNF1B* Deletion	−3	Yes	Kidney Pancreas Genitalia	Yes	HPT Hypospadias
2	29	Μ	*HNF1B* Deletion	−3	Yes	Kidney Pancreas	Yes	Psoriasis Elevated LFT
3	19	Μ	*HNF1B* Deletion	−2.1	Νο	Kidney	No	Elevated LFT
4	15	Μ	p.Asn289Thr	−1.5	Yes	Kidney Pancreas	No	No
5	49	F	p.Asn289Thr	−2.5	Yes	Kidney	No	No
6	5	M	17q12 deletion	NA	Νο	Kidney Genitalia	Yes	Autism cryptorchidism Elevated LFT
7	61	F	p.His336Asp	−1.1	Yes	No	Yes	Ulcerative colitis, HTG

DM: diabetes mellitus; sMg: serum magnesium, eGFR decline rate measured in mL/min/1.73 m^2^; HPT: hyperparathyroidism; LFTs: liver function tests; HTG: hypertriglyceridemia; NA: not available.

## Data Availability

The data that support the findings of this study are available from the corresponding author upon reasonable request.

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
