# Peer review of "The Phenotypic Variability Associated with Hepatocyte Nuclear Factor 1B Genetic Defects Poses Challenges in Both Diagnosis and Therapy"

_ijms, 2024, doi:10.3390/ijms25084552_

Round 1

Reviewer 1 Report

Comments and Suggestions for Authors

In this case report, the authors presented six cases of HNF1B-associated kidney disease exhibiting varying renal and extrarenal manifestations. The authors concluded that this case series emphasized the phenotypic variability associated with HNF-1B related disease, and it underscored that complex clinical presentations may have a retrospectively straightforward explanation through the use of diverse genetic analytical tools.

Comments

This is an interesting case report. The reviewer has some concerns as follows:

1.     Regarding describing the characteristics of individual patients in the text, it is a bit confusing. It is recommended to present the demographic characteristics of patients in a table.

2.     I’m not sure about the description of the results in Table 1. Please explain in more detail and why there are no relevant genetic alterations described in the text?

3.     In Table 1, there are two HNF1A genes. Please check it if there is a mistake.

4.     In line 88, “Despite WE) analysis yielding…” changes to “Despite WES analysis yielding...”

Author Response

Response to Reviewers

Reviewer 1

Ιn this case report, the authors presented six cases of HNF1B-associated kidney disease exhibiting varying renal and extrarenal manifestations. The authors concluded that this case series emphasized the phenotypic variability associated with HNF-1B related disease, and it underscored that complex clinical presentations may have a retrospectively straightforward explanation through the use of diverse genetic analytical tools.

Comments

This is an interesting case report. The reviewer has some concerns as follows:

  1. Regarding describing the characteristics of individual patients in the text, it is a bit confusing. It is recommended to present the demographic characteristics of patients in a table.

Response: We thank the reviewer for the constructive comment. A new table was created describing demographic and main clinical characteristics and it has replaced the previous Table 1.

  1. I’m not sure about the description of the results in Table 1. Please explain in more detail and why there are no relevant genetic alterations described in the text?

Response: The table was designed to illustrate the potential pitfalls of Whole Exome Sequencing (WES) by showing the abundance of data it provides, which can sometimes be misleading. However, we acknowledge that its inclusion may create confusion, and therefore, we have decided to remove it from the revised manuscript. The previous Table-1 has been replaced with a new one summarizing the demographic, clinical, and genetic data of our series.

  1. In Table 1, there are two HNF1A genes. Please check it if there is a mistake.

Response: The table was designed to illustrate the potential pitfalls of Whole Exome Sequencing (WES) by showing the abundance of data it provides, which can sometimes be misleading. However, we acknowledge that its inclusion may create confusion, and therefore, we have decided to remove it from the revised manuscript. The previous Table-1 has been replaced with a new one summarizing the demographic, clinical, and genetic data of our series.

  1. In line 88, “Despite WE) analysis yielding…” changes to “Despite WES analysis yielding...”

Response: The recommended correction has been made

Reviewer 2 Report

Comments and Suggestions for Authors

Q1 Line 75 except for one, newly diagnosed, patient

except for one newly diagnosed patient

Q2 line 88 Dulaglutide

Daily dose and frequency, please add

Q3: line 88 Despite WE)

WES?

Q4 Despite WES analysis yielding no relevant genetic alterations, MLPA identified a heterozygous HNF1B gene deletion

Why MLPA is more sensitive than WES?

Q5 Based on Fig 1 B, do you think that case 1, and 2, case 4 and 5 are de novo mutation?

Definitely not!

Q6: line 153-154 Timely recognition of extrarenal manifestations, particularly diabetes mellitus, holds the potential for initiating therapeutic interventions that can safeguard renal function

Do you think that DM or DM with poor blood sugar control, is the main factor leading to the decline of GFR? Would it be possible that it is the HNF1B that is the factor not only causing the morphologic changes of the kidney (medullary sponge kidney (Please see https://www.ncbi.nlm.nih.gov/pmc/articles/PMC7710890/), but also impairing kidney functions (https://diabetesjournals.org/care/article/40/11/1436/37001/Diabetes-Associated-Clinical-Spectrum-Long-term)?

Author Response

Reviewer 2

Comments and Suggestions for Authors

Q1 Line 75 except for one, newly diagnosed, patient except for one newly diagnosed patient

Response: The recommended correction has been made

Q2 line 88 Dulaglutide Daily dose and frequency, please add

Response: we added “currently at the dose of 1.5mg weekly”

Q3: line 88 Despite WE) WES?

Response: The recommended correction has been made

Q4 Despite WES analysis yielding no relevant genetic alterations, MLPA identified a heterozygous HNF1B gene deletion

Why MLPA is more sensitive than WES?

Response: the sensitivity of MLPA versus WES depends on the specific genetic variation being detected and the regions of the genome being analyzed. MLPA may be more sensitive for detecting CNVs and large variations in specific genomic regions, while WES may be more sensitive for detecting SNVs and indels within the exonic regions. Both techniques have their own strengths and are often used complementary to each other in genetic analysis especially in patients with HNF1b syndrome.

Based on the reasonable reviewer’s statement we have changed this sentence as follows: “Despite WES analysis yielding no relevant genetic alterations (data not shown), MLPA identified a heterozygous HNF1B gene deletion (Figure 1B) confirming that MLPA can more easily identify large deletions compared to WES.”

Q5 Based on Fig 1 B, do you think that case 1, and 2, case 4 and 5 are de novo mutation?

Definitely not!

Response: we agree with the reviewer that these patients do not carry de novo mutations, considering their ancestors had a history of diabetes. We have not claimed though that they have de novo mutations. Only patient 3 has a de novo mutation since his parents were not carriers of the specific mutation.

Q6: line 153-154 Timely recognition of extrarenal manifestations, particularly diabetes mellitus, holds the potential for initiating therapeutic interventions that can safeguard renal function. Do you think that DM or DM with poor blood sugar control, is the main factor leading to the decline of GFR? Would it be possible that it is the HNF1B that is the factor not only causing the morphologic changes of the kidney (medullary sponge kidney (Please see https://www.ncbi.nlm.nih.gov/pmc/articles/PMC7710890/), but also impairing kidney functions (https://diabetesjournals.org/care/article/40/11/1436/37001/Diabetes-Associated-Clinical-Spectrum-Long-term)?

Response: We thank the reviewer for the constructive comment. We have added two paragraphs in the discussion section based on the citations proposed by the reviewer (citations 8 and 9) that actually strengthen our main message: “The HNF1B protein plays a crucial role in regulating tissue-specific gene expression in various epithelial cells found in multiple organs, including the kidney, pancreas, liver, and genitourinary tract. Additionally, transient expression of HNF1B is observed in tissues such as the neural tube, epididymis, seminal vesicles, prostate, and uterus. This diverse expression pattern accounts for the highly variable presentation of HNF1B-related disorders (8).”

“The most common clinical renal presentation observed is chronic tubulointerstitial nephritis, characterized by nonsignificant urinalysis and a gradual decline in renal function. Renal biopsy shows a pattern of interstitial fibrosis and tubular atrophy. Notably, a similar phenotype is often observed in patients with heterozygous variants of UMOD, MUC1, and REN genes, which cause an autosomal dominant form of tubulointerstitial nephritis. Hence, HNF1B nephropathy is now classified as part of the group of disorders termed Autosomal Dominant Tubulointerstitial Kidney Disease (8, 9)”.

A third paragraph addressing the comment about diabetes contribution to CKD progression, was also added, stating: “Timely recognition of extrarenal manifestations, such as diabetes mellitus and hypertension, holds the potential for initiating therapeutic interventions that can delay renal function decline. In a large French study, renal biopsy was performed in 20 patients and revealed various lesions, including interstitial fibrosis in 10 and diabetic glomerular disease in 5. The progression of kidney disease was found to be more strongly associated with the presence of hypertension, a nucleotide variation versus deletion, proteinuria, and age, rather than solely with the presence or absence of diabetes (9). Therefore, all of these factors should be considered when designing our therapeutic strategy.”

Reviewer 3 Report

Comments and Suggestions for Authors

This case series emphasizes the phenotypic variability associated with HNF-1B related disease. The authors specified the fact that in contemporary practice, all patients with renal cysts, genital tract malformations, hypomagnesemia, liver test abnormalities  and diabetes should undergo screening for HNF1B-associated disease.

Some remarks:

1. You wrote that: “Clinical and laboratory data were gathered from the patients' medical records over a 12-year period (except for one, newly diagnosed, patient). Please give details concerning what did you followed in general.

2. Please upload the institutional agreement as a suppl material and at lines 178-79 add the No. of the agreement and the date in which it was approved.

3. All 6 cases are male. Can you explain this aspect? Did you find case reports for women in the literature? can you add them to the discussions?

4. Typos: line 88: WE).

Comments on the Quality of English Language

Minor editing of English language is required.

Author Response

Reviewer 3

This case series emphasizes the phenotypic variability associated with HNF-1B related disease. The authors specified the fact that in contemporary practice, all patients with renal cysts, genital tract malformations, hypomagnesemia, liver test abnormalities and diabetes should undergo screening for HNF1B-associated disease.

Some remarks:

  1. You wrote that: “Clinical and laboratory data were gathered from the patients' medical records over a 12-year period (except for one, newly diagnosed, patient). Please give details concerning what did you followed in general.

Response. A collaborative protocol was established and implemented across three outpatient clinics at our tertiary hospital: adult nephrology, pediatric nephrology, and endocrinology. The aim was to investigate the prevalence and detailed clinical and genetic features of patients suspected to have HNF1B-related syndrome, based on the presence of multicystic/dysplastic kidney disease and/or maturity-onset diabetes of the young. This protocol was developed and approved in 2015. Over the years, we have diagnosed 8 individuals with HNF1B gene alterations, with 7 of them exhibiting pathological clinical features thus far. As previously mentioned, clinical and laboratory data were collected from the patients' medical records spanning a 12-year period (except for one newly diagnosed patient), encompassing both retrospective and prospective information post-diagnosis.

We have now added the relative details in the patients and methods section.

  1. Please upload the institutional agreement as a suppl material and at lines 178-79 add the No. of the agreement and the date in which it was approved.

Response: The study has been approved by the ethics committee of the University Hospital of Heraklion, with a protocol number 2529/17-6-2015. We have now added the relative information in the patients and methods section.

  1. All 6 cases are male. Can you explain this aspect? Did you find case reports for women in the literature? can you add them to the discussions?

Response: Thanks for the interesting comment. There is no gender predilection and there are several reports for women. We actually added a female case in our series (Patient 7) avoiding the need to further comment on this issue. This woman bears the p.His336Asp alteration that was designated as VUS in the ClinVar database, and was not included in our initial submission. However, it was recently recognized as pathogenic in several members of 2 families in Croatia and one family in Turkey, making us to consider the inclusion of this patient.

  1. Typos: line 88: WE).

Response: The recommended correction has been made

Comments on the Quality of English Language. Minor editing of English language is required.

Response: We tried to improve the quality of English Language.

Reviewer 4 Report

Comments and Suggestions for Authors

In this case series, Petrakis and coworkers studied the wide range of clinical manifestations linked with genetic defects of Hepatocyte Nuclear Factor 1B. The state-of-art should be better presented to introduce the case series.

The Reference list should be extended, since 8 literature references are not enough for a case series that should be published in an international journal.

Comments on the Quality of English Language

English language could be improved, but it does not hinder the readability  of the work.

Author Response

Reviewer 4

Comments and Suggestions for Authors

In this case series, Petrakis and coworkers studied the wide range of clinical manifestations linked with genetic defects of Hepatocyte Nuclear Factor 1B. The state-of-art should be better presented to introduce the case series.

The Reference list should be extended, since 8 literature references are not enough for a case series that should be published in an international journal.

Comments on the Quality of English Language. English language could be improved, but it does not hinder the readability of the work.

Response: We thank the reviewer for the constructive comments. We have now expanded the literature, the discussion and the presented cases by adding 3 more citations and one more family with an interesting HNF1B gene alteration. This woman bears the p.His336Asp alteration that was designated as VUS in the ClinVar database, and was not included in our initial submission. However, it was recently recognized as pathogenic in several members of 2 families in Croatia and one family in Turkey, making us to consider the inclusion of this patient.

We also tried to improve the quality of English Language.

Round 2

Reviewer 1 Report

Comments and Suggestions for Authors

This revised manuscript has a great improvement and can be accepted.

Reviewer 4 Report

Comments and Suggestions for Authors

The revised manuscript addressed this reviewer's requests and it has been improved as recommended.